# Steelmaking Process Optimised through a Decision Support System Aided by Self-Learning Machine Learning

Doru Stefan Andreiana [1,*], Luis Enrique Acevedo Galicia [1], Seppo Ollila [2], Carlos Leyva Guerrero [1], Álvaro Ojeda Roldán [1], Fernando Dorado Navas [1] and Alejandro del Real Torres [1]

1   IDENER, IT Department, 41300 Sevilla, Spain; luisenrique.acevedo@idener.es (L.E.A.G.); carlos.leyva@idener.es (C.L.G.); alvaro.ojeda@idener.es (Á.O.R.); fernando.dorado@idener.es (F.D.N.); alejandro.delreal@idener.es (A.d.R.T.)
2   SSAB Europe Oy, Processes Development Steelmaking, 60100 Seinäjoki, Finland; seppo.ollila@ssab.com
*   Correspondence: doru.stefan@idener.es

**Abstract:** This paper presents the application of a reinforcement learning (RL) algorithm, concretely Q-Learning, as the core of a decision support system (DSS) for a steelmaking subprocess, the Composition Adjustment by Sealed Argon-bubbling with Oxygen Blowing (CAS-OB) from the SSAB Raahe steel plant. Since many CAS-OB actions are selected based on operator experience, this research aims to develop a DSS to assist the operator in taking the proper decisions during the process, especially less experienced operators. The DSS is intended to supports the operators in real-time during the process to facilitate their work and optimise the process, improving material and energy efficiency, thus increasing the operation's sustainability. The objective is that the algorithm learns the process based only on raw data from the CAS-OB historical database, and on rewards set according to the objectives. Finally, the DSS was tested and validated by a developer engineer from the CAS-OB steelmaking plant. The results show that the algorithm successfully learns the process, recommending the same actions as those taken by the operator 69.23% of the time. The algorithm also suggests a better option in 30.76% of the remaining cases. Thanks to the DSS, the heat rejection due to wrong composition is reduced by 4%, and temperature accuracy is increased to 83.33%. These improvements resulted in an estimated reduction of 2% in $CO_2$ emissions, 0.5% in energy consumption and 1.5% in costs. Additionally, actions taken based on the operator's experience are incorporated into the DSS knowledge, facilitating the integration of operators with lower experience in the process.

**Keywords:** machine learning; reinforcement learning; Q-learning; steelmaking process CAS-OB; decision-support system; optimisation algorithm

## 1. Introduction

The Composition Adjustment by Sealed Argon-bubbling with Oxygen Blowing (CAS–OB) is a secondary steelmaking process developed in the 1980s by Nippon Steel Corporation [1]. The main goals during this process are homogenisation, temperature control and composition adjustment [2]. As a result, CAS-OB has become one of the relevant buffer stations in the secondary metallurgy of steelmaking thanks to its capability of good chemical composition control, steel homogeneity, and reheating [3]. Furthermore, the process enables the consistent correction of high alloy composition and the reheating of the steel using the exothermic reaction between oxygen and aluminium. The study presented has been done under the MORSE project (Model-based optimisation for efficient use of resources and energy) [4] funded by the European Commission within its Horizon 2020 framework programme. The case under investigation pertains to the SSAB Europe Oy Raahe steel plant with two CAS-OB stations [5], which provided the data for performing the training of the algorithms and then performed the validation of the final developed DSS.

### 1.1. Motivation

In large-scale production, such as steelmaking, even small changes in resource and energy consumption can make a difference. Hence, the steel industry is continuously looking to enhance sustainability, as stated in similar studies concerning aluminium reduction and energy efficiency [6,7].

Additionally, besides the efficiency of the plant, it is crucial to consider the high impact of those processes on the environment [8]. Energy consumption constitutes up to 40% of the cost of steel production [9]. Specific energy consumption at Raahe steelworks in 2016 was 17.8 GJ/t of steel and specific $CO_2$ emission 1650 kg/t of steel. One of the objectives of this study is to reduce those numbers and, in consequence, enhance the efficiency and sustainability of the process.

With that purpose in mind, the first step is to minimise the rejected products, and the second is to provide support to the operators. Many of the actions taken by the operators are based on their own experience, which makes the integration of the new staff into the workforce difficult. Therefore, another objective is to gather that knowledge based on operator experience into the DSS, for the ease of new workers' integration.

These kinds of processes usually lack digitalisation and integration of AI-based techniques. Thanks to the advances in those fields in the last years, the objectives can be achieved by replacing the classic control techniques with novel control techniques strengthened by machine learning algorithms, such as an adaptive controller based on Q-Learning [10]. This can speed up the transformation towards industry 4.0 and all its benefits [11], such as the enhancement of the steelmaking process's efficiency [12] and facilitating the work of the operators while accommodating them to digitalisation [13]. A review of the latest methods and tools for improving the steelmaking process was presented by T. Rotevatn et al. in 2015 [14]. Some of these methods have already been applied and have given excellent results [15], which also served as motivation, and provides evidence that this is the right path.

### 1.2. Methodology and Goals

This research aims to develop a decision support system (DSS) to assist the operator in taking the proper decisions since many CAS-OB actions decisions are made based on operator experience. The DSS includes self-learning ability thanks to the core based on Q-Learning, which has already been stated to give excellent results in similar studies of different metallurgic processes [16]. Furthermore, the algorithm will adapt and learn new features while working, improving its suggestions and achieving the goals defined, thanks to the high impact of the self-learning ability [17].

The guidance is expected to reduce additional corrective actions, reduce duration time and minimise failed operations. Furthermore, the Q-Learning algorithm used as the core of the DSS will learn the process from the historical data of two CAS-OB stations provided by SSAB Europe Oy. This way, it is possible to develop an RL algorithm based only on raw data. The steps followed during this research to achieve the objectives proposed are presented in the following list:

1. **Select the most suitable RL algorithm.** There are many RL algorithms [18], and each one suits a different kind of problem. Therefore, this step involves analysing the available algorithms and selecting the most suitable for the process's features.
2. **Transform CAS-OB process into a Markov Decision Process (MDP).** This task involves an exhaustive analysis of the process, including an analysis of historical data from it. Once done, the process's discretisation is undertaken, defining representative states and actions based on the analysis results.
3. **Train the algorithm.** In this study, the objective is to train an RL algorithm only with experimental data. Therefore, historical data is used from the process. The data will be transformed according to the discretisation set before, and episodes will be defined to "feed" the algorithm.

4.  **User interface (UI).** The next step is to incorporate the DSS on the plant and facilitate access through a user interface.
5.  **Test.** The DSS is tested in the CAS-OB steel plant station by development engineers. The test includes validating the correct behaviour of the DSS and gathering data for performance measurement.

## 2. Reinforcement Learning

### 2.1. Origin

Machine learning (ML) main areas are supervised learning and unsupervised learning [19,20]. Nevertheless, a third area has been gaining popularity in the last decade: Reinforcement Learning (RL). What distinguishes it from the other main areas is that it does not rely on correct behaviour examples, as supervised learning does. On the other hand, unlike supervised learning, its objective is not finding common or hidden patterns on data, neither classify.

Reinforcement learning's [21] origin rests on the idea of learning by interaction. RL algorithms aim to solve a problem by direct interaction with the environment, learning by trial and error, based on the consequences of the actions in terms of rewards. Practising and exploring new ways of doing something produces a wealth of information about each action's effects, and it helps define the best action path for solving the problem. Figure 1 presents a scheme with the RL methodology.

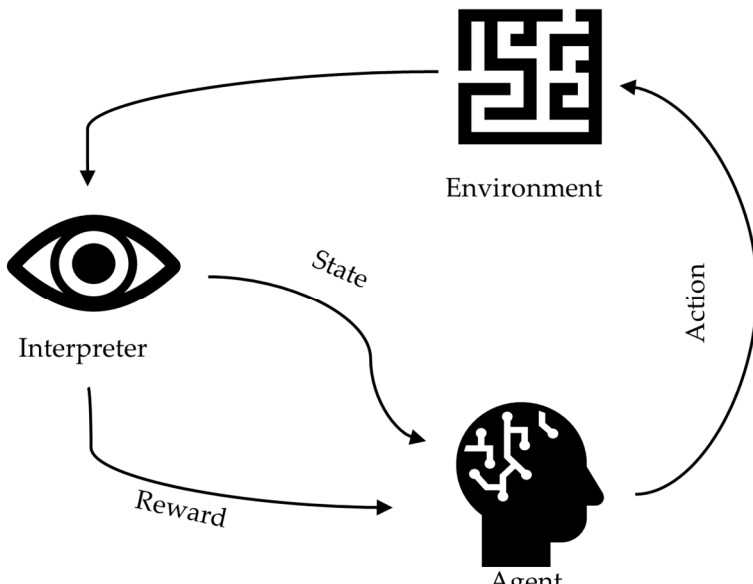

**Figure 1.** Reinforcement Learning methodology.

RL's popularity increased greatly thanks to its application on games defining agents capable of surpassing human abilities, such as Go [22] and Dota 2 [23]. Nevertheless, the RL algorithm can go beyond winning games [24]. Nowadays, thanks to the wide variety of algorithms developed, the applic ation area has greatly expanded due to its advantages over classic control techniques. The classic control techniques and RL algorithms are frequently compared [25].

### 2.1.1. Reinforcement Learning Elements

In Reinforcement Learning, the main elements are the agent and the environment, which interact with each other. However, the interaction involves some other subelements [21]:

-   **Agent.** The learner and decision-maker.

- **Environment.** What the agent interacts with, comprising everything outside the agent. The interpreter defines the state of the environment and gives rewards as feedback to the agent.
- **State (S).** A sample of the environment's features involved in the case under study. It defines the situation of the environment and works as feedback.
- **Action (A).** A control signal of the environment. The defined actions must be those that change the states involved in the problem.
- **Episode.** A succession of steps, composed of the state of the environment and the action taken by the agent. In the case under study, the step ends when the action changes the environment status. At the end of the step, a reward is received based on the new state achieved. These episodes are used to train the RL agent. The format of an episode can be seen in Figure 2.
- **Policy (π).** The core of the RL agent, which defines how the agent behaves in each state, based on what has been learned as the best action. It can be a simple function or a lookup table, although it can involve extensive computation depending on the problem's complexity.
- **Reward (R).** The reward defines the goal of the problem. Each state gets a reward as feedback on how well the RL agent did when taking the previous actions. The RL agent will define a policy maximising the rewards obtained. Therefore, the rewards must be set according to the objectives of the process. Rewards assignment is a critical task because agent behaviour will rely on them.
- **Value function.** There are two types, value-functions and action-value functions. The difference is that the first one calculates the values of the states only, and the second one calculates the values for pairs of state and action. Moreover, in both cases, the formulas have modifications depending on the algorithm. The algorithm applied in this study, Q-Learning, uses an action-value function. The values calculated determine the best action, considering long-term consequences.

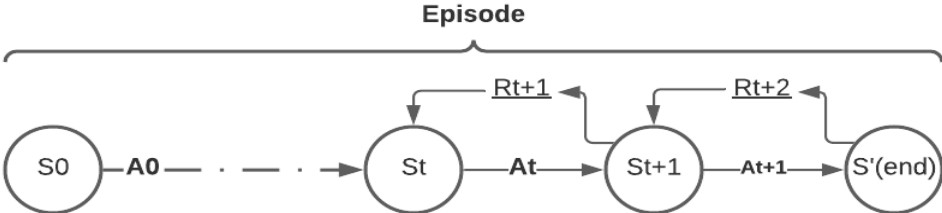

**Figure 2.** Episode structure.

The way the value function is updated is what distinguishes one RL method from another. It is done by following an update rule which depends on the RL method because each one estimates the new values differently. The generic update rule is formulated as in Equation (1):

$$NewEstimation \leftarrow OldEstimation + StepSize[Target - OldEstimation] \qquad (1)$$

where *Target* is the estimated accumulative reward in each state, and the expression [*Target* − *OldEstimate*] is an error in the estimation. Moreover, the variable *StepSize*, known as alpha($\alpha$), controls how much the agent learns about its experience. This parameter is between 0 and 1, and it may be constant or variable.

2.1.2. Markov Decision Process

The Markov decision process is a classical definition of sequential decision-making, where actions influence immediate rewards and subsequent states, and therefore those future rewards. It is a substantial abstraction of the problem of goal-directed learning from interaction. It proposes that any problem can be reduced to three signals that provide communication between the agent and the environment. This concept is used as a mathematical

form of the reinforcement learning problem [26]. However, it is used in many other fields, such as medicine [27] and robotics [28], among many others.

Aiming to frame the problem of learning from interaction, MPDs are the base and a requirement of most of the RL algorithms, amongst them the Q-Learning, used in this case study. In the MDP framework, the learner and decision-maker are called *agents*, and what they interact with is called the *environment*. The interaction is done through *actions*, and the environment responds with its new *state*, and the consequences, namely *rewards*, are set according to the goal. Further on, the agent will aim to maximise the *rewards* through its *actions* and the knowledge learned. Figure 3 illustrates the structure of an MDP.

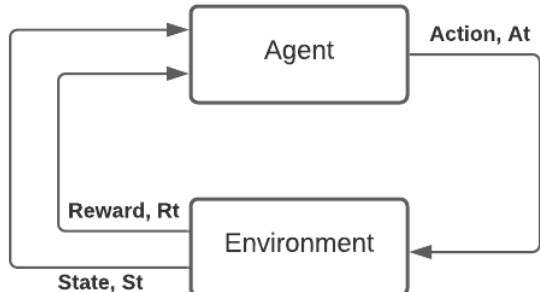

**Figure 3.** Agent–environment interaction in a Markov decision process.

The boundary between agent and environment represents the point beyond which the agent lacks absolute control and knowledge. However, the border does not have to be physical; it can be located at different places depending on the goal. This boundary is defined once the states, actions and rewards are defined. Then, the interaction is divided into *steps* (*t*), and the sequence generated depends on the actions selected and the dynamics of the environment.

### 2.2. Reinforcement Learning Algorithms

RL methods have the advantage that they can learn from direct experience of the process. If the system's operations are saved on a database, these can be used as training data. This also brings up probably the most significant advantage of the RL, continuous training. Moreover, this learning allows the agent to adapt to changes in the process without designing the method again.

The most relevant tabular methods have been studied to find the most suitable RL method for the problem. These methods are briefly described below. Their description also helps clarify the method used.

- Dynamic programming (DP) [29]: this is an assemblage of algorithms capable of computing optimal policies if a perfect environment model is given. This requirement, together with the tremendous computational cost, limits the utility of DP algorithms.
- Monte Carlo (MC) methods [30,31]: these methods are an improvement compared to dynamic programming because their computational expenses are lower, and a perfect model of the environment is not necessary. This last feature means that knowing the environment's dynamics is not required to attain the optimal policy; the agent learns from direct interaction. Instead, these methods only need a simulated or real experience of this interaction in the form of state-action-reward sequences. Another distinctive feature of these classes of techniques is that they estimate values functions by averaging sample returns. Thus, MC methods are very suitable for episodic tasks, since they base their estimations on the final outcome and not on intermediary outcomes.

*2.3. Self-Learning Algorithm: Q–Learning*

Temporal difference (TD) learning is the most significant thread in RL [32]. It is a combination of DP and MC. TD methods, just like MC, are model-free, and additionally, they can learn from raw experience [33]. On the other hand, they assimilate with DP on how the estimation is updated. The values are updated based on previously known estimations, performing the calculation on the fly. Consequently, in contrast with MC methods, TD methods do not have to wait until the end of the episode to determine the value states. TD methods only need to wait until the next time step, and the update is done in "transition". This is why TD methods are so popular. There is no need to know how the episode will end; the agent learns from its estimations. If the estimation were very wrong, it would learn much more. Further on, convergence is still guaranteed. Moreover, TD methods have been proved to be the fastest.

The Q–Learning [34] method is one of the most known and used methods of RL. It is a model-free method and uses an off-policy approach. A side effect of being model-free and a temporal difference method is the capacity of being used online. It can be used online because it is unnecessary to know the episode's ending to update the values. This feature is essential in the problem studied because, in the final version, the agent should be online and recommend actions in real-time during the steelmaking process, adapting to each situation. Besides, this method also can handle stochastic problems, as in the problem treated in this project.

Moreover, Q-Learning is a control method, so it is based on an action-value function, which suits perfectly the problem in hand. The function used approximates directly to the optimal action-value function. Furthermore, it converges faster because it always aims for taking the best action. The duration will primarily be determined by the size of the state and action space. The action-value function used in Q-Learning is shown in Equation (2). It is used to calculate the value of each pair of states and actions. Afterwards, based on these values, the most suitable action, the one with higher value is selected.

$$Q(S_t, A_t) = Q(S_t, A_t) + \alpha \left[ \underbrace{R_{t+1} + \gamma \max_a Q(S_{t+1}, a)}_{Target} - Q(S_t, A_t) \right] \quad (2)$$

- The sub-index $t$ indicates the step during the episodes.
- $\alpha \in (0, 1]$, the step size or learning rate determines to what extent newly acquired information overrides old information. The value could vary during the training, so we learn a large amount at the beginning, and at the end, when the optimal values are reached, there is no override.
- $\gamma \in [0, 1]$, the discount rate, indicates the importance of the next reward, or in this case, the value of the next possible state and action pair.

The pseudocode of Q-learning algorithm is presented in the Algorithm 1 [35]:

---

**Algorithm 1.** Q-learning (off-policy) for estimating $\pi \approx \pi_*$

---

Algorithm parameters: step size, $\alpha \, \epsilon \, (0, 1]$, $\gamma \, \epsilon \, (0, 1]$
Initialise Q (s, a) for all $s \in S^+$, a $\in A(s)$, arbitrarily except that Q (terminal, ) = 0
Loop for each episode:
Initialise S
Loop for each step of the episode:
Choose A from S using policy derived from Q
Take action A, observe R, $S'$

$$Q(S_t, A_t) = Q(S_t, A_t) + \alpha[R_{t+1} + \gamma * max_a Q(S_{t+1}, a) - Q(S_t, A_t)]$$

$S \leftarrow S'$
Until S is terminal

---

### 2.4. Comparison with Classic Control Techniques

Classic control efficiency is mainly limited by the model reliability. The control method is designed to control the model. Moreover, the control design will not adapt to changes in the process. Any modification on the process means defining a new model and developing a new controller.

Meanwhile, RL algorithms, such as Q-Learning, fill the gap limitations of classical control. The algorithm's design only requires knowledge about the process, the states that define it, the actions (control signals), and the consequences of these actions. So, it is helpful to have a model; however, it is not a requirement. Figure 4 presents a summarised scheme of the comparison.

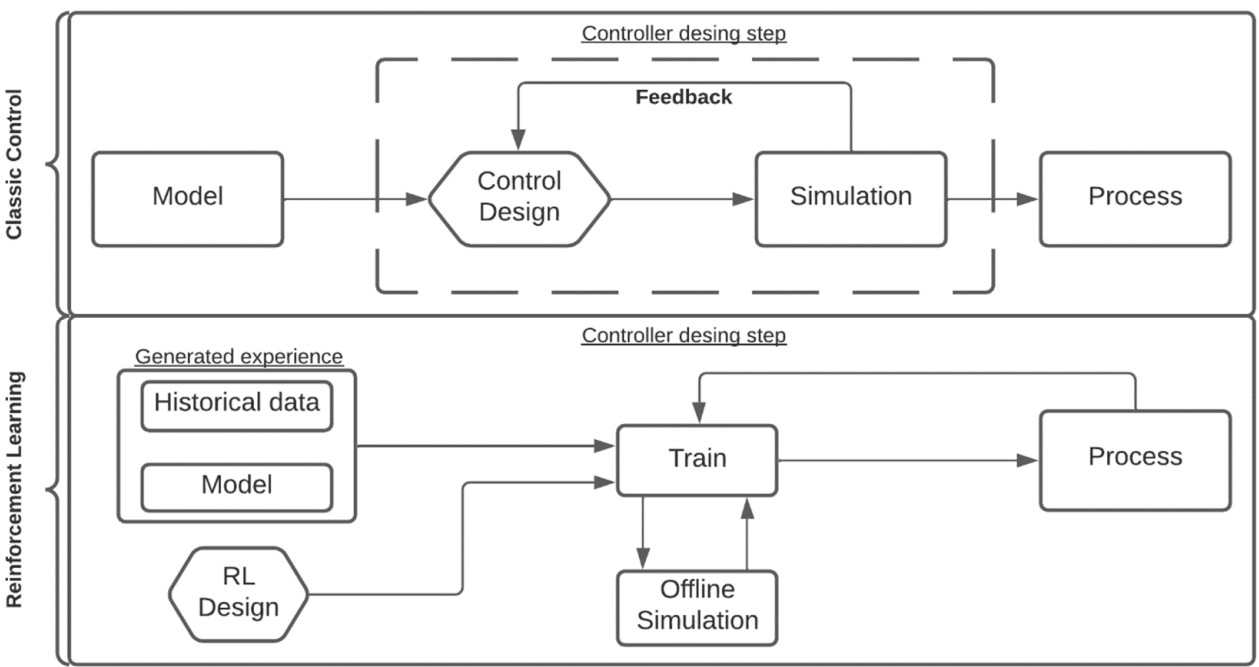

**Figure 4.** Reinforcement learning comparasion with classic control.

### 3. Development of the Solution

As already mentioned, the RL agent will use a Q-Learning algorithm. As its features have already been explained, this section will focus on the actual implementation. Figure 5 shows an overview of the whole developed and integrated solution.

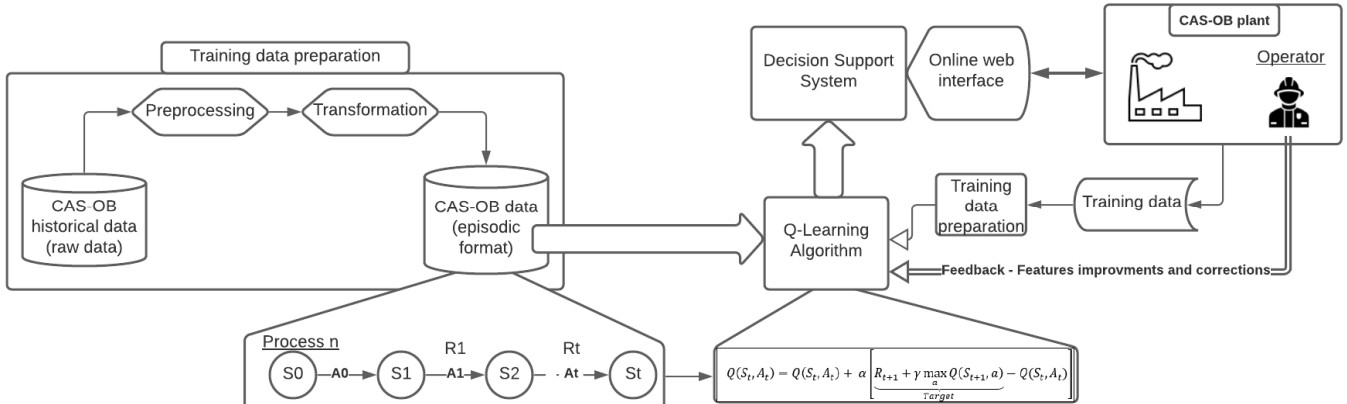

**Figure 5.** Scheme of solution developed and implemented.

In the first place, the historical data from the CAS-OB process must be processed, selecting only the valuable data since the database includes all the information regarding the CAS-OB process. Once the variables needed for the episode definition and the further training are designated, the next step is the transformation into an episodic format. As already explained, the episodic format is necessary due to the MDP nature of the self-learning algorithm. The algorithm receives the state of the environment as an input, and instead of following an action based on the policy, it selects the action defined in the episode. In other words, during the training, the agent will strictly follow the episodes while calculating the values of each state-action pair.

Once the algorithm finishes the training with the historical data and performs some validations, it can be introduced into the DSS implemented in the CAS-OB plant and used by a developer engineer. During this interaction, the algorithm still learns from the data gathered while it is used. Moreover, developer engineer feedback is crucial at the beginning for features improvements, corrections of the behaviour or states and action discretisation.

### 3.1. CAS-OB Steelmaking Process Description

CAS-OB works with liquid steel from a previous process, the *Basic Oxygen Furnace* (BOF) [36]. The main goals for this secondary metallurgy process are homogenisation, temperature control and adjustment of composition. After that, the liquid steel is transferred to the Continuous Casting Machine (CCM) (see Figure 6).

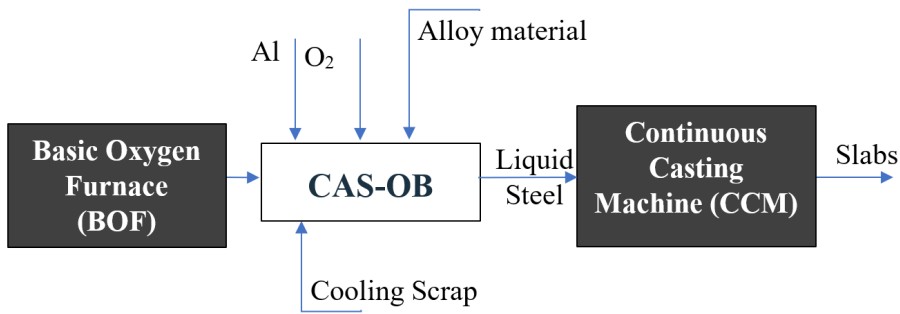

**Figure 6.** Diagram of the steelmaking process summarised.

Composition adjustment starts in the previous phase, the BOF tapping. Afterwards, during the CAS-OB process, the homogenisation and adjustment of the composition of molten steel must be maintained while the temperature is adjusted. Composition adjustment is achieved by adding a stoichiometric proportion of alloys which is calculated using a composition sample.

Temperature and dissolved oxygen measurements are made before alloying in order to verify whether these two variables need to be adjusted. Should there be a corrective action to be taken, it has to be carried out before alloying, in order to ensure no extra oxidation is needed, as it would alter the composition and therefore a subsequent adjustment would be needed. Once this alloying is done, composition is deemed to be satisfactory and no more alloying is necessary, unless a subsequent sample analysis may indicate the need. Liquid steel temperature can be increased by adding aluminium with oxygen blowing, using the exothermic reaction between them for reheating. Whenever refrigeration is needed, cooling scrap is added to decrease the temperature. Likewise, this technique can be used as well for minor temperature adjustments at the end of the process. Considering steel cleanliness, the time at which cooling scrap is added is not so critical, the reason for this being that it does not change the composition significantly, and therefore it can be added whenever it is needed. Once the goal is achieved, the liquid steel moves forward to the third process, the Continuous Casting Machine (CCM).

### 3.2. Definition of Environment

The process will be transformed into an MDP. This requires defining a finite representative number of states and actions. The definition of these states and actions is based on the historical data analysis and the process's objectives.

#### 3.2.1. Historical Data Analysis and Treatment

The historical data includes all the information from a real CAS-OB plant, gathering 9720 steel treatments, also called "heats". Only the significant information for the formulation of the problem was extracted from this data, such as the measurements, targets and actions taken by the operators. Table 1 presents the parameters gathered from the historical data.

**Table 1.** Data gathered from CAS-OB historical data.

| Treatment Data | Measurements | Actions | Targets |
|---|---|---|---|
| Heat number | Measuring time | Addition time | Composition |
| Start of treatment | Composition, % of each alloy | Material | Temperature |
| End of treatment | Temperature | Quantity, kg | |
| Start of reheating | Dissolved oxygen | | |
| Reheating time, s | | | |
| End of reheating | | | |
| Count of reheating | | | |
| Total aluminium, including reheating, kg | | | |
| Cooling scrap, kg | | | |
| Steel amount | | | |
| The time when treatment should be ready | | | |

Making use of the knowledge on how the plant works and the historical data provided by the steel factory the process was modelled as an MDP. The data was previously treated eliminating inconsistencies, outliers and corrupted information, and afterwards the states and actions were properly defined.

The heat number classifies the steel treatments in the database. Each heat contains information about measurements, actions, targets and what has been done during the process. Those are the most important information for the classification of the states and the actions. Timestamps will help set up the state and action transition chronologically and generate a complete episode when something has been done.

If some information is missing on some heat, e.g., missing essential measurement or misleading information, the first option will be to fill the gap if possible to preserve the episode and not lose data. As an example, there are instances in which steel temperature is recorded to be at 0 °C. This is obviously an impossible value for the process; as a result on these occasions temperature was removed from the log, but not the other variables. The missing values of temperature would be inferred manually afterwards.

After cleaning and analysing the data, the classification and definition of states can be done. The current database consists of 13,800 heats in total. The duration of the episodes varies depending on how many times reheating has been done. In addition, the data provide information about the features of the measurements, target, and actions, e.g., "adding Al".

The final dataset is divided based on how many times reheating has been done. This division is done because depending on the reheats done, the data will be treated differently. The distribution is presented in Figure 7. The idea of the RL is to generalise. Focusing excessively on exceptional cases would guide the agent too much. Hence it will not find new optimised ways of solving the problem. It must discover by itself those exceptional cases.

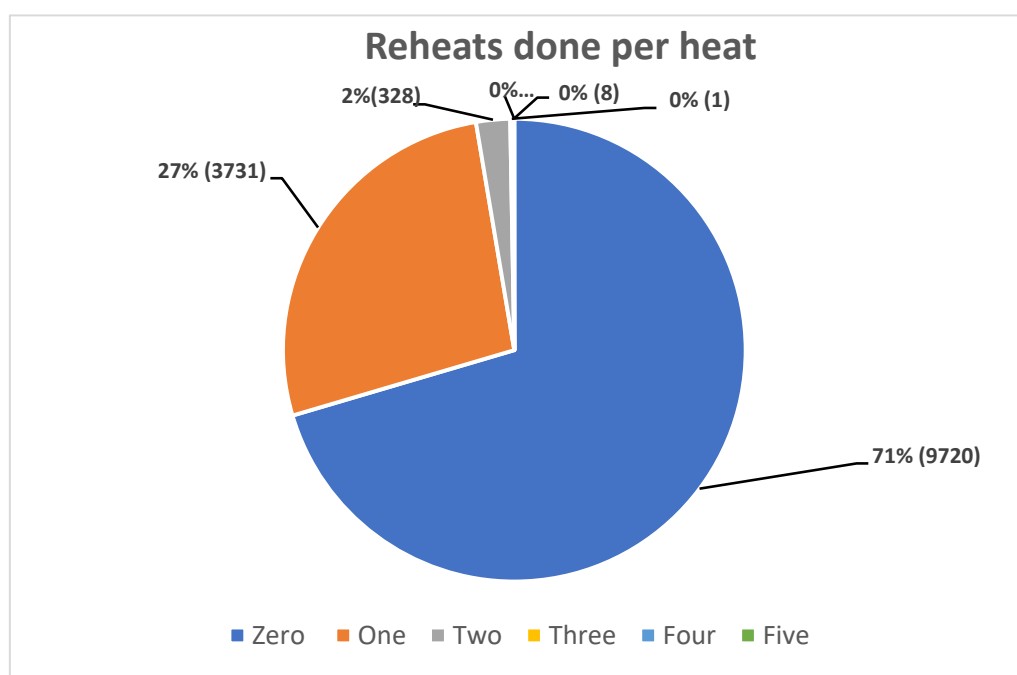

**Figure 7.** Pie chart representing the number of episodes per number of times reheating was done during the episode.

3.2.2. States

The states for the CAS-OB use case are defined by five parameters: the dissolved oxygen, the composition status, composition grade, steel mass, and the temperature difference. These parameters are continuous. Henceforth they must be discretised. If not, the state space will have infinite size. Below are listed and detailed the discretisation done to all the parameters.

- Mass of the steel:

The mass of steel is characterised into two categories, high and low. Its value for most of the steel heats is around 125 tons. Nevertheless, there are heats with reduced mass. The mass for these types of steel is 15–20% below the usual value. This must be taken into account because if the mass is considerably lower than usual, then the amounts of Al and O2 needed to heat the steel will be lower. A smaller mass requires reduced actions for the same results. So, the process with liquid steel mass below 110 tonnes will be considered low, and higher values will be regarded as high.

- Dissolved oxygen (DO):

DO is measured in ppm, and its target value is as close to zero as possible to ensure steel cleanliness. After the BOF process, the dissolved oxygen content is typically several hundred ppm. Dissolved oxygen is decreased by adding aluminium into steel first during BOF tapping and then during the CAS-OB process if needed, according to dissolved oxygen measurements. Aluminium reacts with oxygen forming aluminium oxide. Ideally, the exact amount of aluminium required for the reaction should be added.

Nevertheless, sometimes the amount added is not precise, or the reaction did not go on exactly as expected. From the historical data, the mean calculated value is equal to 5.18 ppm. The discretisation of the DO reduces the possible values to five ranges, all close to the mean. The discretisation of the DO is presented in Table 2.

- Temperature.

This state is defined as the difference between the target temperature during the process and the measured temperature. The discretisation is shown in Table 3 and is done

based on the temperature difference distribution at the end of the process, considering any waiting time due to the scheduled ending time.

**Table 2.** Dissolved oxygen discretisation.

| Dissolved Oxygen Intervals | Codification of the Intervals |
|:---:|:---:|
| DO < 1 | 0 |
| $1 \leq DO < 2$ | 1 |
| $2 \leq DO < 3.5$ | 2 |
| $3.5 \leq DO < 5$ | 3 |
| $5 \leq DO < 7$ | 4 |
| $7 \leq DO < 10$ | 5 |
| $10 \leq DO$ | 6 |

**Table 3.** Difference temperature states.

| Temperature Difference Intervals | Codification of the Intervals |
|:---:|:---:|
| $\Delta T < -20$ | $-3$ |
| $-20 \leq \Delta T < -10$ | $-2$ |
| $-10 \leq \Delta T < -2$ | $-1$ |
| $-2 \leq \Delta T \leq 2$ | 0 |
| $2 < \Delta T < 10$ | 1 |
| $10 \leq \Delta T < 20$ | 2 |
| $20 \leq \Delta T < 30$ | 3 |
| $30 \leq \Delta T < 40$ | 4 |
| $40 \leq \Delta T$ | 5 |

- Composition state.

The composition of the steel processed includes 16 elements: C, Si, Mn, P, S, Al, Nb, N, V, Ni, Ti, B, Ca, Cu, Cr, and Mo. The measurements taken during the process show the percentage of each element in the liquid steel, These are compared with the goals for those percentages, namely, minimum, target and/or maximum. Ideally, each percentage should reach its target if that is provided. However, if this value is not provided, then the composition should fit between the maximum and minimum percentage levels by setting the mean of both as the target percentage. The aimed result is to achieve the type of steel that is required.

Until the measurement is done, the composition may be considered incorrect. Once the measurements are taken, if any of the percentages are below the target, the composition is erroneous, and alloying is necessary., Otherwise, the composition is correct, and alloying is not necessary. So basically, there are **two possible states, correct and incorrect composition**. However, the incorrect composition state is disaggregated in four states: **Very Close, Close, Far and Very Far**. However, while the composition is incorrect, the action will always be "alloying", and how far the composition is from the target does not alter that. However, a higher resolution is required to estimate the temperature decrease accurately. The further the composition is from the target, the more alloys will be added, which could result in a higher drop in temperature. So, by making that division, the RL agent will learn that the alloying can reduce the temperature.

- Composition grade.

The grade of steel provides more information about the steel treated and its features. Some grades require different actions for the same consequences. Hence, the grades have been categorised into four groups, each with steels of different features.

3.2.3. Actions

CAS-OB process is complex and involves many actions. However, the RL agent will take into account only the actions, which are mainly based on operator experience and

whose timing, and the amounts, directly affect the process's outcome. All the actions are independent from one another, and there can be one taken per state. The list below details the actions considered and how they have been discretised:

- Addition of aluminium:

  Aluminium is usually added to reduce the dissolved oxygen in the liquid steel or adjust the composition during the process. Nevertheless, it is also added besides O2 to increase the temperature, which is called reheating. However, only about 25% of all CAS-OB heats are reheated. The reheating serves to increase the temperature. Despite added aluminium, blown oxygen can also react with other alloying elements, consequently spoiling the composition. Therefore, reheating should always be performed before correcting the composition.

  The RL agent recommends when to add aluminium to reduce the dissolved oxygen and when to add it for reheating. Additionally, it suggests the amount in kilograms of aluminium that should be added. To recommend the amount of aluminium, this must be discretised to facilitate the estimation. To do so, it is necessary to know the usual amount of aluminium added during the process to make those adjustments. For that purpose, the historical data from the process was analysed. Figure 8 shows how many processes (episodes) there were for each amount of Al that has been added. It is seen that the usual quantity is a multiple of ten, and the bigger it is, the fewer times it is added.

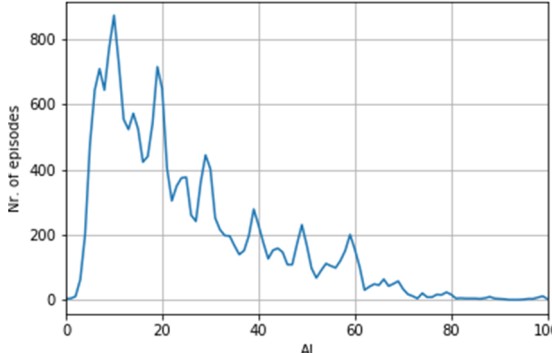

**Figure 8.** Number of episodes x Kg of Al that has been added.

- To perform the same analysis for the aluminium added for reheating, some pre-processing of the data was needed since the amount added for reheating is not specified in the historical data. Fortunately, the total amount of Al added during the process is specified. Hence, by subtracting the Al not added for reheating, the Al for reheating is obtained. The results of the analysis are similar to the previous analysis. The only difference is that the interval of amounts added is larger, and more significant amounts are usually added. Lastly, the RL agent does not specify the amount of $O_2$ added with the Al for the exothermic reaction, because the operator system directly calculates it.

  A total of 26 discretised actions (15 for reheating, 11 for DO reduction) regarding the addition of aluminium will be considered, either for reheating or dissolved oxygen (Table 4).

**Table 4.** Aluminum addition discretisation.

| Aluminium for Reheating | Aluminium for DO Correction |
| --- | --- |
| $O_2$ + Al 150 kg | Al 100 Kg |
| Between 135 and 145 kg of Al + $O_2$ | Al between 85 and 95 Kg |
| Between 125 and 135 kg of Al + $O_2$ | Al between 75 and 85 Kg |
| . . . | . . . |
| Between 15 and 25 kg of Al + $O_2$ | Al between 6 and 15 Kg |
| Less than 15 kg of Al + $O_2$ | Al less than 6 Kg |

- Cooling Scrap:

This action is used to reduce the temperature of the steel when needed. It is essential to mention that the RL agent may recommend cooling down the steel and ending the process faster. Nevertheless, the operator can refuse it if there is plenty of time for the steel to cool down by itself. The operator calculates the cooling scrap amount based on the temperature decrement desired.

- Alloying:

This action corrects the composition and drastically reduces the temperature of the steel. The current control system of the plant provides the amount of alloying the process needs, and this action indicates to the operator the best time to add the alloys calculated by this control system.

### 3.2.4. Problem Size

Considering all the combinations between the sub-states of the parameters defined, the **size of the state space is 2520**. In other words, there are 2520 possible states. The size is quite large; however, many states will hardly ever occur if the physical nature of the problem is taken into consideration. On the other hand, summing all the possible actions, considering the discretisation of each one, there are **28 possible actions**. To conclude, this results in a Q table of 2520 × 28 size, hence **70.560 Q-values**. Many Q-values will probably not be calculated. Nevertheless, that is not a problem for the optimisation problem because, as mentioned with the states, many combinations of states and actions would make no sense in the real process, and just as the operator would do, the RL agent will directly discard those options.

### 3.2.5. Rewards

As explained before, rewards are the feedback for the agent indicating how well it is performing. This step is essential, as the behaviour of the agent will be based on how these rewards are assigned. The rewards must define the goal of the process correctly to achieve proper training and, in consequence, the correct functioning of the RL. If rewards are not appropriately set, the agent will not learn correctly how to optimise the process. Rewards must represent the goal of the process, hence the objective of the RL. As explained before, the RL will try to achieve the maximum value reward. With all this in mind, the rewards must be placed beneficially for the goal.

A reward will be set for all the possible end states, which in this case, means all of them as every state could be an end state. The reward will be selected depending on the end temperature, DO and composition status. Ideally, the temperature should be equal or very close to the target, DO approximately zero and the composition correct. Therefore, the reward for the ending status which such characteristics will be set with a maximum value, and rewards for approximate states with lower values. An incorrect composition will be determined directly as a bad ending, independently of the other states, and the RL agent will be penalised. The rewards will be placed as presented in Figure 9.

If the process ends with incorrect compositions, the "reward" is −100. Thus, almost all the rewards have their opposing pair for a negative end. By doing this, the categorisation and calculation of the values are more effective. Besides, for each incorrect action taken, there is a minor penalty (negative reward). The consequence of that constant minor negative reward will train the RL agent to finish faster and achieve the objectives in the minimum number of actions. Moreover, this penalty varies depending on the state. For example, the penalty is bigger if the temperature is much lower than the target, consequently, the RL agent will prioritise heating.

To summarise, the rewards will be set as follows:

- Maximum reward for the final states on the goal
- Intermediate reward for final states close to the goal, for example, good composition and temperature but a high dissolved oxygen level.

- High negative reward if the end composition is wrong.
- Intermediate negative reward for correct composition but wrong temperature or high DO.
- A small negative reward for each action taken to favour ending faster and reducing the number of actions taken.

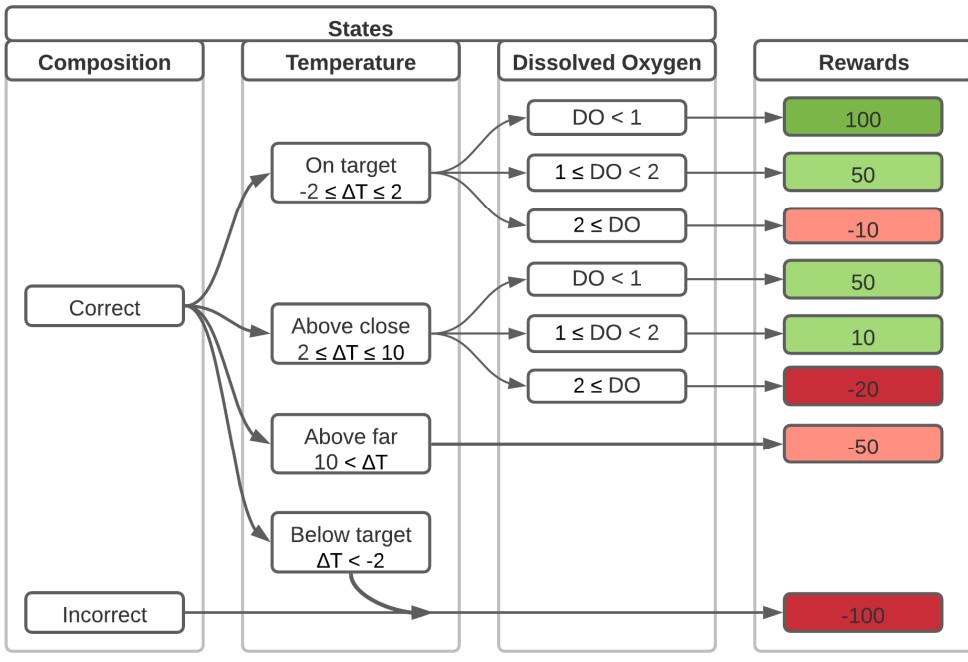

**Figure 9.** Rewards for the final states with the correct composition.

## 4. Implementation and Validation

### 4.1. Training Data

The training of the algorithm is done through the historical data of the process by SSAB. However, the data is not discretised and is not in the required format to be used as training data for the Q-learning algorithm. The first step is the discretisation of the data in state and actions. How is done has already been detailed above. The second step, detailed in this section, is the transformation of the data into episodic format and with the structure needed by the Q-learning formula (Figure 2).

Each episode will correspond to a heat, which is already classified. Regarding the discretisation, the data is transformed into the states and actions defined before. Once done each state must be associated with the action taken taken in that state, unless it is the ending state. The procedure followed is explained through an example of a real heat. After the discretisation, the format of the states of the heat is shown in Table 5 and of the actions in Table 6. However, some adjustments are required.

Firstly, the composition status is not entirely correct. In the process, once the alloying is performed, it is assumed that the composition is right and no further measurements of the composition are made, unless if the operator decides they are needed. Consequently, the composition status is not updated and, thus, the state is always "Close", hence incorrect composition. This first problem is solved by checking when the alloying action has been performed and uploading all the composition states that follow it to "Good".

Secondly, if we compare the addition times with the measuring times, it can be observed that several actions are taken between two state updates (those are marked on orange in the tables). Even if that can be done in practice, the configuration of the RL algorithm proposed does not allow taking several actions at the same time. Therefore, those states must be separated accordingly, associating an action to each one, so that they can be used as training data for the Q-learning algorithm. In order to correct this, the consequence of each action will be considered, in other words, how each action between those

measurements would change the state. Based on that, the state will be divided, as many times as actions taken, and its substates will be updated according to the consequences of the actions, previously identified. In the example shown in the tables, the actions taken were "Al" and "Cooling Scrap". The purpose of these actions was to reduce the DO and the temperature. Therefore, the state is divided into two and is updated sequentially, as if the actions had been performed separately.

**Table 5.** Example of heat's state format after discretisation.

| Measuring Time | T | T Difference | DO | Comp. | STATE |
|---|---|---|---|---|---|
| 01-01-2019 01:32:53 | $40 \leq \Delta T$ | 79 | $5 \leq O2 < 10$ | Close | S0 |
| 01-01-2019 01:42:29 | $40 \leq \Delta T$ | 48 | $5 \leq O2 < 10$ | Close | S1 |
| 01-01-2019 01:51:15 | $20 \leq \Delta T < 30$ | 23 | $3.5 \leq O2 < 5$ | Close | S2 |
| 01-01-2019 01:56:13 | $10 \leq \Delta T < 20$ | 14 | $3.5 \leq O2 < 5$ | Close | S3 |
| 01-01-2019 01:59:53 | $2 \leq \Delta T < 10$ | 4 | $O2 < 3.5$ | Close | S4 |

**Table 6.** Example of heat's actions after discretisation.

| Addition Time | Material | Amount, kg | Action |
|---|---|---|---|
| 01-01-2019 01:33:45 | Al | 39 | Al between 35 and 45 kg |
| 01-01-2019 01:33:45 | Cooling Scrap | - | Cooling Scrap |
| 01-01-2019 01:36:45 | Cooling Scrap | - | Cooling Scrap |
| 01-01-2019 01:44:54 | Al | 11 | Al between 6 and 15 kg |
| 01-01-2019 01:44:54 | Cooling Scrap | - | Cooling Scrap |
| 01-01-2019 01:47:42 | Alloying | - | Alloying |
| 01-01-2019 01:57:02 | Al | 10 | Al between 6 and 15 kg |
| 01-01-2019 01:57:02 | Cooling Scrap | - | Cooling Scrap |

Thirdly, there are also states without actions, intermediate measures done to check the status and if the actions taken have achieved their purpose. If this problem overlaps with the previous problem presented, the information gathered from the additional measurements is used as additional support in the procedure followed in the previous problem.

Finally, sometimes, although not frequently, there are actions taken after the last measurement. In these cases, the end state is estimated in function of the consequences of the action taken.

The example presented does not include a reheating sequence. However, same problems may occur and same procedure is followed. It is only necessary to add the reheating action between the already defined actions. The addition is done by comparing the sample times with the reheating start time.

In conclusion, all these problems leave gaps in the episode which must be filled. Once the gaps are filled, episodes with MDP structure are available and ready to be used as training data for the algorithm. The result of the procedures is presented in the following section, because they also serve as check on the work done so far.

Validation of Environment Transformation

The objective of this task was to validate if the CAS-OB process has been correctly transformed into an MDP and, hence, to validate the training data transformation. This validation is crucial since the agent's training is pointless if the data does not represent CAS-OB's process. To achieve the validation, the fact that every process gathered from the CAS-OB historical data was correctly transformed into an episodic format was checked.

Table 7 below includes the final result of heat transformation into MDP format. Therefore, the data can be used for training.

**Table 7.** Final result of episodes generation.

| ΔT | DO | Comp. | Action | |
|---|---|---|---|---|
| | | Close | Al between 35 and 45 kg | $S_0-A_{01}$ |
| $40 \leq \Delta T$ | $5 \leq O2 < 10$ | Close | Cooling Scrap | $S_{01}-A_{02}$ |
| | | Close | Al between 6 and 15 kg | $S_1-A_{11}$ |
| | $3.5 \leq O2 < 5$ | Close | Cooling Scrap | $S_{11}-A_{12}$ |
| $20 \leq \Delta T < 30$ | $3.5 \leq O2 < 5$ | Close | Alloying | $S_2-A_{13}$ |
| $10 \leq \Delta T < 20$ | $3.5 \leq O2 < 5$ | Good | Al between 6 and 15 kg | $S_3-A_{31}$ |
| | $O2 < 3.5$ | Good | Cooling Scrap | $S_{31}-A_{32}$ |
| $2 \leq \Delta T < 10$ | $O2 < 3.5$ | Good | F | $S_4$ |

*4.2. Training and Validation of the Algorithm*

Once the environment is validated, the next step is to validate the algorithm. RL algorithms are validated by achieving the convergence of the policy during the training stage. The policy of the Q-Learning algorithm converges if the Q-values remain constant between different trainings. The convergence proves that there is a solution for solving the problem. Hence the rewards are coherent with the environment, and the algorithm suits the problem.

The Q-Learning algorithm's training consists of following the historical data repeatedly until the solution is found, or in other words, it converges. The convergence is checked by calculating the maximum difference between the values from the previous training and the new ones. If the subtract is lower than the established accuracy parameter, set at 0.001, the convergence is achieved. Figure 10 illustrates the progress during the training, and the red line represents the accuracy parameter. Convergence is completed in 89 iterations.

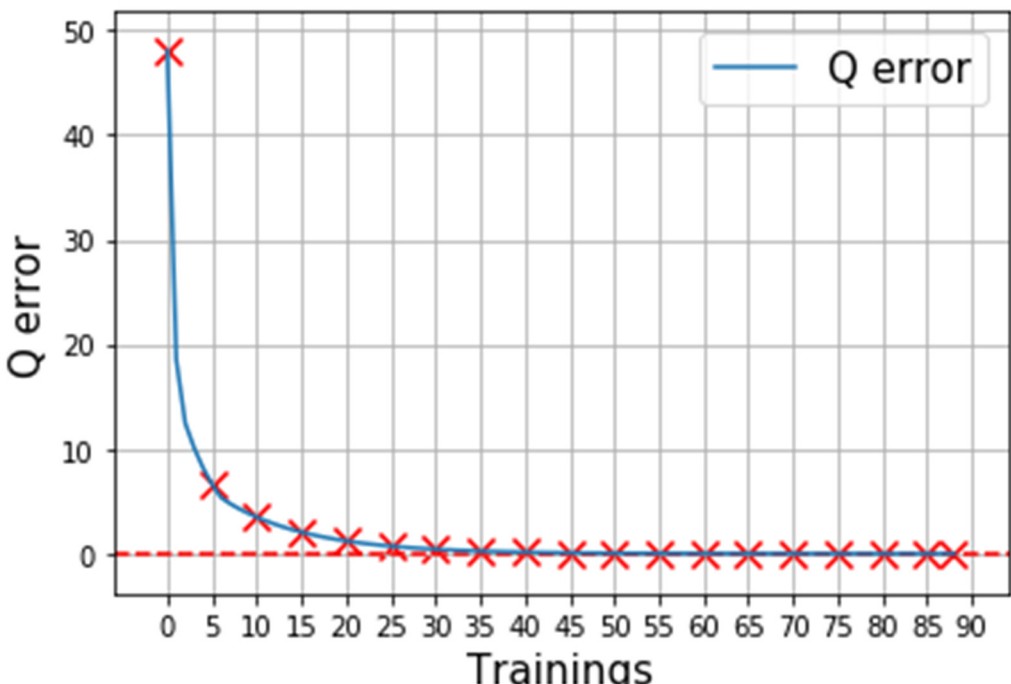

**Figure 10.** Training convergence.

Once the training ends, the program retains the Q values in a CSV file. Afterwards, the file is loaded and used to recommend actions. Moreover, it keeps learning and adjusts the values with the new experience gained. The training with the historical data is an initialisation of the Q values. Therefore, the DSS will recommend the right actions from the beginning while learning and enhancing its performance.

Nevertheless, before implementing the algorithm in the online DSS interacting with the real process, the policy must be validated. To accomplish this, the policy was validated by simulating episodes from the historical record following its recommendations. The validation continued by checking if the action recommended matched or was similar to the action taken in the historical dataset. The validation concludes when the DSS recommends coherent actions.

*4.3. Online Interface*

The online interface aims to facilitate the operator's work and to support him in real-time during the process operation. In Figure 11 the interface is shown. The RL UI will read from an OPC server the necessary data to define the state of the process. Therefore, it will not be necessary to set the inputs manually (although it is still possible to modify any input manually). Once the state is defined, the recommended action is shown, and every time the state changes, the action will change accordingly.

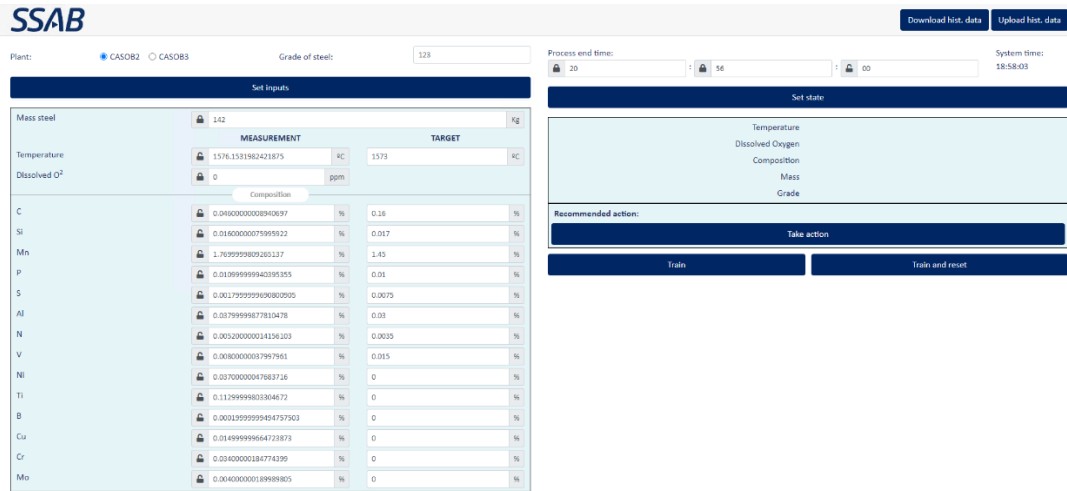

**Figure 11.** Decision support system online interface installed in CAS-OB process.

## 5. Testing and Results

The testing of the DSS and the gathering of results for performance measurement has been performed during two months at the CAS-OB plant. At the beginning of the testing, the focus was on the functionalities offered by the DSS and finding possible bugs. Thanks to the development engineers' feedback, it was possible to enhance the DSS by making the following adjustments to the environment definition and the learning algorithm:

- Adjust the rewards to be stricter on agent learning.
- Correction on composition status calculation.
- Addition of constant states during the process, such as the grade and mass of the steel.
- Higher resolution for dissolved oxygen states and aluminium addition actions.

In the case of the dissolved oxygen, some additional states closer to the target were added, which resulted in a better performance recommending aluminium to reduce the dissolved oxygen. Concerning the aluminium addition, the maximum amount that the system can suggest was increased. Before the change, the maximum amount that could be recommended was 50 Kg.

Thanks to these changes, the support offered by the DSS was more precise and helpful for the process. Additionally, it was observed that the RL agent prioritises firstly the composition and secondly the temperature, in worst scenarios aiming at least to maintain the temperature above target, which fits perfectly with the goals of the process. Further on, during all the testing periods, data has been gathered to measure the DSS's performance.

First, all the states processed by the DSS are stored together with the recommended action. Afterwards, these actions are compared with the actions taken by the operator, available in the historical data. By doing this, it was possible to analyse in detail the policy of the Q-learning algorithm. The analysis reflected that 69.23% of the time, the action concurred with the action taken by the operator.

Nevertheless, the 30.76% left does not mean the action recommended was incorrect, but the operators preferred to follow their experience. However, analysing the end of those heats, it was observed that many of them do not end ideally. In other words, the actions taken by the operator were not the best. Additionally, in order to ensure if the recommended action was better, a process engineer of CAS-OB process from SSAB, analysed the actions recommended and compared them with the actions taken by the operator. Estimating the consequences of the actions with the support of an external model, the process engineer from SSAB concluded that following the actions recommended, the heat would have ended perfectly in most of the cases. However, since CAS-OB is such a complex process and the DSS is a novel tool for the operator, it is understandable that experienced operators would follow their own experience instead of the recommendation of a tool as novel as the proposed DSS.

Finally, the efficiency of the process is measured with the following key performance indicators (KPIs) presented in Table 8:

**Table 8.** Key performance indicators for the RL agent application to the CAS-OB process.

| KPI | Description | Units |
|---|---|---|
| Heat rejections | The percentage of heat rejections due to the wrong composition | % |
| Temperature hitting rate of CAS-OB heats | The percentage of heats in CAS-OB with the correct temperature in continuous casting | % |

Analysing the episodes generated with the RL agent recommendations during the testing period, the heat rejections are reduced by **4%**. On the other side, **83.33%** of the heats end with the temperature on target or slightly above. The second KPI is expected to increase, at least to 89% over time with the continuous training of the agent. Additionally, both results, mainly the rejections reduction, reduced raw materials and energy consumption ($\sim-0.5\%$) and, hence, the $CO_2$ emissions ($\sim-2\%$) and costs ($\sim-1.5\%$). The estimated percentages may seem low but given the huge scale of the steel production in the CAS-OB plant of SSAB, the impact is high. Those estimations were done by SSAB based on their internal data concerning their production, comparing results before and after the integration of the DSS.

## 6. Discussion

To conclude, it has been proved that a DSS using the Q-Learning algorithm can learn a complex process such as the steelmaking process CAS-OB. Moreover, it was possible to train an RL algorithm only with raw data from the historical database. Afterwards, it was validated and tested in the CAS-OB plant with direct interaction with the process. Additionally, thanks to the self-learning ability of the algorithm used as the core of the DSS, it will keep training, adapting and improving its performance while it is used.

The RL algorithm has learned how the process works, and the DSS can support the operator in real-time. Additionally, with constant interaction with the real process, the agent will learn every detail and improve its performance. In the future, it may even be possible to operate automatically while being supervised by the operator. Finally, with this research, the advantages of RL have been demonstrated. Moreover, it remains clear that RL algorithms, such as Q-Learning, are potent tools capable of learning and solving even complex steelmaking processes. This statement encourages further investigation in this direction and application in other steelmaking processes, thus opening a new way of enhancing the performance and sustainability of the steelmaking industry.

**Author Contributions:** D.S.A.: Conceptualisation, formal analysis, software, methodology and writing—original draft; L.E.A.G.: supervision, review and editing; S.O.: validation and review; C.L.G.: conceptualisation, supervision and review, project administration; A.d.R.T.: conceptualisation, funding acquisition; F.D.N.: conceptualisation, supervision and review; Á.O.R.: conceptualisation. All the authors have contributed to this manuscript. All authors have read and agreed to the published version of the manuscript.

**Funding:** This research has been conducted under the MORSE project, which has received funding from the European Union's Horizon 2020 research and innovation program under grant agreement No. 768652.

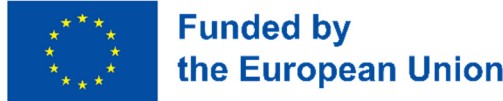

**Institutional Review Board Statement:** Not applicable.

**Informed Consent Statement:** Not applicable.

**Data Availability Statement:** Not applicable.

**Conflicts of Interest:** The authors declare no conflict of interest.

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
