# Peer review of "Steelmaking Process Optimised through a Decision Support System Aided by Self-Learning Machine Learning"

_processes, doi:10.3390/pr10030434_

Round 1

Reviewer 1 Report

The author proposes the subject of optimizing the steelmaking process through a self-learning machine learning-assisted decision support system. This subject is mainly aimed at the CAS-OB process, and aims to develop a decision support system to help operators make correct decisions in the process. Especially inexperienced operators, ultimately improve the smelting efficiency. But the article has the following problems:

  1. The title of the table should be at the top of the table.
  2. The flow chart can be drawn more beautifully.
  3. The format of some parts of the article can be modified appropriately.
  4. How to prove that the final result of the recommended operation is correct when the recommended operation does not match the actual.
  5. At the conclusion, whether the improvement of the algorithm to the entire smelting process can be shown concretely, such as how much the energy consumption is reduced, and how much the CO2 emission is reduced.
  6. There is less data to support the conclusion. It is suggested to reflect the data in a more intuitive way to improve the credibility of the conclusion.

Author Response

1. The title of the table should be at the top of the table.

Changed.

2. The flow chart can be drawn more beautifully.

Figure 4 has been updated, supposing that was that could be improved. However, if anyone else should be updated let me know. 

3. The format of some parts of the article can be modified appropriately.

There have been performed few changes in order to improve that (see in file attached). 

4. How to prove that the final result of the recommended operation is correct when the recommended operation does not match the actual.

The process engineer of the CAS-OB process, who led the testing activities of the DSS, analysed all those cases. Thanks to its knowledge about the process and an external model, the engineer estimated what would have been the results would have been if the recommended actions had been followed. 

5. At the conclusion, whether the improvement of the algorithm to the entire smelting process can be shown concretely, such as how much the energy consumption is reduced, and how much the CO2 emission is reduced.

It has been added an estimation in percentage reduction of the CO2 emissions, costs and energy consumption. It can be found in the conclusion section and in the abstract.

6. There is less data to support the conclusion. It is suggested to reflect the data in a more intuitive way to improve the credibility of the conclusion.

Few changes have been performed in the text. Check the file attached.

Please see the attachment to check all the changes done.

Reviewer 2 Report

The manuscript proposes the application of a reinforcement learning (RL) algorithm, concretely Q-Learning, as the core of a decision support system (DSS) for a steelmaking subprocess, the Composition Adjustment by Sealed Argon-bubbling with Oxygen Blowing (CAS-OB) from the SSAB Raahe steel plant. The manuscript clearly shows the modeling process of Q-Learning algorithm in steelmaking subprocess, including the introduction of reinforcement Learning, the corresponding states, actions and rewards, training and verification as well as the improvement of performance of the combination of reinforcement Learning and steelmaking subprocess. These findings are of great significance for promoting reinforcement learning as a decision-making aid in industrial production. In addition, the manuscript has the following problems and needs to be revised.

  1. This manuscript introduces five parameters of State and three types of actions in Section 3.2.3 and Section 3.2.2. It is suggested to further describe Action space, State space and Q table.
  2. In Section 3.2.3, the action of ‘Addition to aluminium’ is divided into 26 actions. Is there a combination relationship with Cooling Scrap and Alloying actions, or is it composed of 28 separate actions?
  3. In Section 3.2.4, Figure 9 shows the reward in the termination state. In Line 499-500, the part also describes ‘Besides, for each action taken, there is a minor penalty.’ The reward and penalty suggest further introduction, so as to improve the model's reward mechanism.
  4. In section 4.1, the training process is still not clear. More detailed programming process should be given, including but not limited to mathematical formula steps and pseudo-code writing.
  5. In Section 5, the author adopts two key performance indicators (KPIs): Heat rejections and Temperature hitting rate of CAS-OB heats, which is relatively interesting. But the part also describes ‘Both results, mainly the rejections reduction, reduce raw materials and energy consumption, hence the CO2 emissions.’, a simple calculation of the level of reduction is recommended.
  6. Some related works are suggested to be cited, such as:

(1) Water Res. 189 (2021) 116576 , DOI: 10.1016/j.watres.2020.116576;

(2) Chemosphere, 234 (2019) 893-901. DOI: 10.1016/j.chemosphere.2019.06.103

Author Response

1. This manuscript introduces five parameters of State and three types of actions in Section 3.2.3 and Section 3.2.2. It is suggested to further describe Action space, State space and Q table.

Done. It has been added an additional sub-section (3.2.4) before "Rewards" sub-section. In this new section is summarised the size of the problem, state and action space and Q table size.

2. In Section 3.2.3, the action of ‘Addition to aluminium’ is divided into 26 actions. Is there a combination relationship with Cooling Scrap and Alloying actions, or is it composed of 28 separate actions?

It is composed of 28 separate actions. I have clarified it in the Action section introduction and in the new section added. 

3. In Section 3.2.4, Figure 9 shows the reward in the termination state. In Line 499-500, the part also describes ‘Besides, for each action taken, there is a minor penalty.’ The reward and penalty suggest further introduction, so as to improve the model's reward mechanism.

There have been added some more details aiming to clarify that. Such as what that penalty means and how it affects the learning of the agent.

4. In section 4.1, the training process is still not clear. More detailed programming process should be given, including but not limited to mathematical formula steps and pseudo-code writing.

Almost the whole section has been updated presenting clearly the problems that may occur and how have been solved. 

5. In Section 5, the author adopts two key performance indicators (KPIs): Heat rejections and Temperature hitting rate of CAS-OB heats, which is relatively interesting. But the part also describes ‘Both results, mainly the rejections reduction, reduce raw materials and energy consumption, hence the CO2 emissions.’, a simple calculation of the level of reduction is recommended.

It has been done an estimation in percentage reduction. Added in the results section and also abstract.

6. Some related works are suggested to be cited.

Done. Both have been cited.

Please see the attachment to check all the changes done.

Reviewer 3 Report

This paper presents the application of a reinforcement learning (RL) algorithm, concretely Q-Learning, as the core of a decision support system (DSS) for a steelmaking subprocess, the Composition Adjustment by Sealed Argon-bubbling with Oxygen Blowing (CAS-OB) from the SSAB Raahe steel plant.

The references in this paper are quite outdated. Almost all of them are more than years ago. It is strongly suggested for the authors to add more updated references, to provide the existing examples in the academic and in industry, and to highlight the differences between the proposed system and the existing examples.

The results presented are overall of a good standard. However, the problem statement and motivation could be stronger or more clearly highlighted.

It seems there are some points in the paper are unclear, such as the conditions imposed to develop the main results, and some computation issues. The author needs to carefully look at those and make some corrections or explanations.

Author Response

1. The references in this paper are quite outdated. Almost all of them are more than years ago. It is strongly suggested for the authors to add more updated references, to provide the existing examples in the academic and in industry, and to highlight the differences between the proposed system and the existing. examples.

There have been added some more references from the last years.

2. The results presented are overall of a good standard. However, the problem statement and motivation could be stronger or more clearly highlighted.

Several changes have been performed on that in order to present it more clearly. Also, some figures and tables have been updated. The new references added should contribute to that too.

3. It seems there are some points in the paper are unclear, such as the conditions imposed to develop the main results, and some computation issues. The author needs to carefully look at those and make some corrections or explanations.

There have been performed several corrections and changes on that in order to detail that and present it more clearly.